# Caffeic Acid and Diseases—Mechanisms of Action

**DOI:** 10.3390/ijms24010588

**Published:** 2022-12-29

**Authors:** Nela Pavlíková

**Affiliations:** Division of Cell and Molecular Biology & Center for Research of Diabetes, Metabolism, and Nutrition, Third Faculty of Medicine, Charles University, 100 00 Prague, Czech Republic; nela.pavlikova@lf3.cuni.cz; Tel.: +420-26-710-2657; Fax: +420-26-710-2650

**Keywords:** caffeic acid, cancer, diabetes, obesity, atherosclerosis, Alzheimer’s disease

## Abstract

Caffeic acid belongs to the polyphenol compounds we consume daily, often in the form of coffee. Even though it is less explored than caffeic acid phenethyl ester, it still has many positive effects on human health. Caffeic acid can affect cancer, diabetes, atherosclerosis, Alzheimer’s disease, or bacterial and viral infections. This review focuses on the molecular mechanisms of how caffeic acid achieves its effects.

## 1. Introduction

When we drink coffee (even the one without caffeine) or red wine, we consume a molecule with highly diverse and interesting effects on our health: a natural polyphenolic compound called caffeic acid. A heavy coffee drinker can consume up to 500 mg of caffeic acid per day; people who do not drink coffee consume up to 25 mg of caffeic acid [1]. However, coffees (and red wine) are not the only sources of caffeic acid in our diet. Many other plant products contain caffeic acid, including apples, plums, lingonberries, black chokeberries, and many herbs of the mint family, e.g., sage, thyme, oregano, marjoram, oregano, or spearmint [1]. Black chokeberries seem to be the most potent source of caffeic acid (645 mg/100 g of dry weight). In comparison, the caffeic acid content in coffee ranges from 9 to 14 mg/100 g [2] or up to 87 mg/100 g, according to [1]. Other sources of caffeic acid are its naturally occurring esters: chlorogenic acid [3], rosmarinic acid [4], and caffeic acid phenethyl ester [5].

Unlike the information about caffeic acid content in various food, data about caffeic acid plasma levels in humans are scarce. It seems that both caffeic acid absorption and metabolism are fast [6], and 1 h after consuming 300 mL of red wine, the caffeic acid level reached a concentration of 28 nM [7]. 

The structure of caffeic acid (aromatic core, conjugated double bond, and hydroxyl groups) allows it to function as an antioxidant, but its effects are far from limited only to that. The published data show effects on various types of cancers, diabetes, obesity, and neurodegenerative diseases like Alzheimer’s or Parkinson’s. This review focuses on the mechanisms of those caffeic acid effects.

## 2. Caffeic Acid as an Antioxidant

The antioxidant effects of caffeic acid play an essential role in many beneficial effects on human health. Khan and coworkers summarized the antioxidant effects of caffeic acid against various types of free radicals supremely [8]. Therefore, we will mention the antioxidant (and prooxidant) effects of caffeic acid only briefly.

Caffeic acid consists of an aromatic core substituted in position 1 with an unsaturated three-carbon chain containing a carboxylic group and in positions 4 and 5 with two hydroxyl groups. It belongs to the so-called hydroxycinnamic acid group: aromatic acids with a C6–C3 skeleton. Caffeic acid’s structure represents an effective trap for radicals; the combination of an aromatic core with a conjugated side chain (Figure 1) allows for an easy delocalization of unpaired electrons. By giving hydrogen to quench the radicals, caffeic acid serves as a primary antioxidant [9]. The hydroxyl group in the *para*position towards the side chain stabilizes free electrons even better. Another way how caffeic acid works as an antioxidant is by chelating the metals with its two hydroxyl groups. Metal ions decompose peroxide into free radicals. By preventing them from doing it, caffeic acid functions as a secondary antioxidant [9].

Nevertheless, the chelating ability of caffeic acid is also responsible for its occasional pro-oxidant ability. After chelating Cu^2+^, the Cu^2+^ can be reduced to Cu^+^. That leads to a cascade of reactions, which produce, among others, superoxide radicals and hydroxyl radicals [10]. A large amount of endogenous copper in the human body occurs in, e.g., lymphocytes. Therefore, the combination of caffeic acid and endogenous copper ions can result in oxidative damage, e.g., DNA breaks [11].

Caffeic acid also prevents the formation of reactive oxygen species (ROS) by inhibiting 5-lipoxygenase. This enzyme turns arachidonic acid into leukotrienes and participates in forming ROS [12].

## 3. Caffeic Acid and Cancer

Multiple studies exist that describe the antiproliferative effect of caffeic acid against various types of cancer cells. Caffeic acid can affect cancer cells alone or in combination with anticancer drugs, which could decrease the anticancer drug dose or help prevent or overcome resistance against those drugs. 

### 3.1. Cancer Prevention

Cooking meat, especially well-done meat, forms heterocyclic amines [13], compounds that act as mutagens and carcinogens [14]. Caffeic acid can inhibit the synthesis of some of them, e.g., PhIP (2-amino-1-methyl-6-phenyl-imidazo[4,5-b]pyridine), which occurs in heated protein-rich food [15]. Caffeic acid probably reacts with phenylacetaldehyde, an intermediate product in PhIP synthesis [16]. Caffeic acid also increased the efflux of PhIP into the intestine lumen by upregulation of ABC transporters p-glycoprotein and breast cancer resistance protein (BCRP) in the apical membrane of the intestine cells [17].

### 3.2. Liver Cancer

Liver cancer is the sixth most common cancer in the world [18], and hepatocellular carcinoma represents the most-diagnosed type among liver cancer cases [19]. Besides chronic hepatitis B and hepatitis C virus infection, its risk factors also include obesity, tobacco, and alcohol usage, and its incidence is generally increasing [19]. One reason for this cancer type’s relatively high morality is that hepatocellular carcinoma responds poorly to treatment due to its high vascularization [20]. The key molecule responsible for angiogenesis in hepatocellular carcinoma cells is a vascular endothelial growth factor (VEGF);both hypoxia-inducible factor 1α (HIF-1α) and pro-inflammatory NF-κB upregulate the expression of VEGF [21,22]. In hepatocellular carcinoma HepG2 and HCC97Hcells, caffeic acid (20 µM) reducedJNK-1-mediated stabilization of HIF-1α and, in this way, decreased the level of active HIF-1α available [21]. In HepG2 cells, caffeic acid (100 µM) inhibited the activity of NF-κB/IL-6/STAT3 signaling, which decreased the expression of VEGF [23]. It also inhibited another downstream product of NF-κB: matrix metalloproteinase 9 (MM-9), which promotes tumor invasiveness and metastases [20,24]. By reducing the expression of both VEGF and MM-9, caffeic acid acted as a potent anti-tumor agent against hepatocellular carcinoma cells. According to Yang and coworkers [25], caffeic acid (20 μM) also decreased the expression of mortalin(mitochondrial 70 kDa heat shock protein), which is an upstream inducer of PI3kB, NF-kB, and VEGF signaling. They observed those effects in three hepatocellular cell lines (HepG2 cells, Hep3Bcells, and sorafenib-resistant HuH7 cells). In hepatocellular carcinoma WCH-17A cells, a higher concentration of caffeic acid(1 mM) blocked proliferation and induced apoptosis by disrupting mitochondrial potential [26]. In rat hepatoma N1-S1 cells, caffeic acid (1 mM) inhibited lactate efflux and, in this way, decreased the effectiveness of anaerobic metabolism [27]. Concerning in vivo experiments, in rats with hepatocellular carcinoma induced by diethylnitrosamine, caffeic acid (100 mg/kg) reduced the histopathological changes and normalized levels of alanine transaminase (ALT), aspartate aminotransferase (AST), alkaline phosphatase (ALP), total bile acid, total cholesterol, HDL and LDL [28]. To summarize, the primary way how caffeic acid affects hepatocellular carcinoma in vitro is inhibiting VEGF expression and upstream pathways (Table 1); in vivo it positively affects hepatic function and reduces histopathological changes.

### 3.3. Breast Cancer

The effects of caffeic acid on breast cancer cells are less described than those of caffeic acid phenethyl ester, and the information about their mechanism is scarce. Breast cancer is the most commonly diagnosed cancer in women [29] and the most common cancer overall [30,31]. The presence or absence of estrogen receptors, progesterone receptors, and receptors for an endothelial growth factor (HER2) plays a significant role in breast cancer therapy and survival [32].

In ER-positive breast cancer cell line MCF7, caffeic acid acted as an antiestrogen [33]; it downregulated the expression of estrogen receptor (ER), insulin-like growth factor 1 (IGF-1) receptor, and the level of activated PKB/Akt kinase, as well as suppressed the growth of cells. ER, IGF1 receptor and PKB/Akt participate in growth regulation pathways in estrogen-sensitive breast cancer cells [33]. In the ER-negative breast cancer cell line MDA-MB-231, the effect of caffeic acid was less prominent [33]. The same study also associated a moderate to high consumption of coffee with a lower breast cancer invasiveness in vivo [33]. In another study using ER-positive MCF7 cells, caffeic acid (171 μg/mL) stimulated the expression of the p21 gene (CDKN1A) [34]; the protein product of this gene arrests the cell cycle. Nevertheless, caffeic acid also stimulated the gene expression of a gene encoding anti-apoptotic protein MCL1 (myeloid leukemia cell differentiation protein) [34], which is not desirable when treating cancer cells. In a triple-negative MDA-MB-231 breast adenocarcinoma line, caffeic acid (50 μM)decreased the migration ability of cancer cells [35,36].

To summarize, caffeic acid inhibits estrogen receptor expression and PKB/Akt signaling in ER-positive cell lines (Table 1); in ER-negative cell lines, the antiproliferative effect is less prominent.

### 3.4. Skin Cancer

The major risk for skin cancer development is the skin’s exposure to UV light. Besides melanomas, skin cancers include non-melanoma skin cancers, e.g., basal cell carcinoma or squamous cell carcinoma.

Caffeic acid protects the skin against cancer on multiple fronts (Figure 2). In the skin, UV light forms ROS that can break the sugar–phosphate spine of DNA [29]. By scavenging ROS, caffeic acid protects DNA against breakage [37]. UV light also forms thymidine dimers in the DNA strand; to repair thymidine dimers, the cell employs a repairing mechanism called nuclear excision repair [29]. In human dermal fibroblasts and mouse skin, caffeic acid (40 μM) prevented the UVB-induced the loss of proteins necessary for nuclear excision repair: xeroderma pigmentosum protein C (XPC), general transcription factor IIH subunit (TFIIH-p44), xeroderma pigmentosum protein A (XPA), and excision repair cross-complementation group 1 (ERCC1), as well as the loss of PTEN [37]. PTEN inhibits the PI3K/Akt signaling pathway, which is often constitutively active in skin cancer cells due to mutations. Additionally, PTEN is necessary for nuclear excision repair [38]. In Swiss albino mice, the pretreatment with caffeic acid (15 mg/kg) prevented UVB light-induced inflammation. Caffeic acid decreased tumor necrosis factor alpha (TNF-α), interleukin-6 (IL-6), cyclooxygenase-2 (COX-2), and NF-κB levels in the exposed mice, possibly by inhibiting the expression of peroxisome proliferator-activated receptor gamma (PPARγ) [39].

In squamous cell carcinoma induced in mice by chronic UVB irradiation, caffeic acid (15 mg/kg) downregulated the expression of inducible nitric oxide synthase (iNOS) and vascular endothelial growth factor (VEGF), upregulated p53expression, and reduced tumor growth [39]. In A431 skin cancer cells, SK-MEL-5 melanoma cells, and SK-MEL-28 melanoma cells, caffeic acid (40 μM directly inhibited ERK1/2 activity and, in this way, disrupted the MAP kinase signaling pathway that promotes tumor growth [40]. Caffeic acid significantly decreased the cell viability of cutaneous melanoma cell line SK-Mel-28 in the same doses that significantly increased the viability of the non-cancer cell line [41]. Caffeic acid also prevented the endothelial growth factor (EGF)-induced neoplastic transformation of human keratinocyte HaCat cells [40].

In transformed human keratinocyte HaCaT cells, caffeic acid decreased the activity of the NF-kB/Snail signaling pathway [42]. Snail inhibits E-cadherin; therefore, Snail inhibition promotes the migratory ability of cancer cells, i.e., metastases [43].

To summarize, caffeic acid can inhibit the PI3K/Akt, MAPK, and NF-kB signaling pathways in skin cancer cells (Table 1), decrease inflammation and oxidative stress and keep nuclear excision repair functional due to stimulation of PTEN expression.

### 3.5. Lung Cancer

Lung cancer is the most common cancer in men and the second-most common cancer in women [44]. Lung cancers include two main groups: non-small cell lung carcinoma and small cell lung carcinoma, which is more aggressive. The data concerning the effect of caffeic acid on lung cancer is controversial. Caffeic acid (600 μM) decreased the viability of human non-small-cell lung cancer H1229 cells but not control cells (human bronchial epithelium non-cancer cells) [45]. In H1299 cells, co-exposure to caffeic acid (100 μM) and cytostatic paclitaxel (10 μM) inhibited cell proliferation more than paclitaxel alone [45]. The co-exposure increased the expression of the pro-apoptotic proteins Bid and Bax, caspase-3/7 and 9 activity, and the expression of 6hosphor-JNK and 6hosphor-ERK1/2 in both H1299 cells and H1299-xenografts in nude mice [45]. Increased levels of phosphorylated p-JNK and p-ERK1/2 would typically represent bad news because the MAPK pathway canonically stimulates cell proliferation. Nevertheless, in some cancer types, activated JNK inhibits aerobic glycolysis and supports apoptosis [46]. According to Lin and coworkers [47], the co-treatment of H1299 cells with paclitaxel and 100 μM caffeic acid increased the viability of H1299 cells (paclitaxel concentration was not disclosed). The caffeic acid exposure also increased the expression of the anti-apoptotic proteins survivin and Bcl-2 in another non-small cell lung cancer cell line, A549 [47]. Nevertheless, in mouse lung adenocarcinoma LA-795 cells, caffeic acid (60 μM) decreased the cell viability to approximately 50%. It also decreased the protein expression of phospho-MEK1/2, phospho-ERK1/2 (members of MAPKinase pathway), cyclin D, beta-catenin (promoters of cell proliferation), and vimentin (a marker of epithelial-mesenchymal transition) [48]. The authors identified the inhibition of the calcium-activated chloride channel TMEM16A, a channel with multiple roles in cancer [49], as the primary mechanism behind those changes. In the mouse xenograft, caffeic acid (5.4 mg/kg) combined with doxorubicin (4.1 mg/kg) significantly decreased the size of tumors [48].

To summarize, most (but not all) data described the antiproliferative effect of caffeic acid against lung cancer; the mechanism often includes an alteration of the MAPK signaling pathway (Table 1).

### 3.6. Oral Cancer

Alcohol and tobacco consumption represents the major risk factors for this less prevalent type of cancer. The most common cancers of the oral cavity and pharynx are head and neck squamous cell carcinomas (HNSCC) [29,50].

Low concentrations of ethanol (2.5–10 mM) increased the growth and migration activity of oral squamous cell carcinoma cells [51]; caffeic acid (50 and 100 μM) reversed the effect. The same authors [52] described that caffeic acid (50 and 100 μM) decreased the viability of the human head and neck squamous carcinoma cells (HNSCC) line (Detroit 562) due to cell cycle arrest in G0/G1 phase. In human tongue squamous cell carcinoma cells (CAL-27), caffeic acid (65 μg/mL) decreased the cell viability while increasing the protein expression of p53, a protein able to promote cell cycle arrest and apoptosis [53]. It also increased the protein expression of proline dehydrogenase/proline oxidase (Table 1), a major enzyme that degrades proline in cells [53]. An increased proline level in cancer cells is connected with a poorer prognosis [54].

### 3.7. Cervical Cancer

Cervical cancer is the fourth most common cancer in the world for women, with the incidence higher in countries with lower incomes [55]. The primary cause of this type of cancer is infection with human papillomavirus (HPV) [55]. Several publications have described a positive effect of caffeic acid against this type of cancer cells.

A combination of cisplatin (11 μM) and caffeic acid (300 μM) significantly increased apoptosis in cervical cancer cell lines HeLa (HPV-18-positive), SiHa, and CaSki (HPV-16-positive), and C33A (HPV-negative) when compared to cisplatin itself [56]. In the non-cancerous VERO cell line, neither cisplatin nor caffeic acid nor their combination significantly increased the number of apoptotic cells [56]. Tyszka-Czochara and coworkers published three articles describing the effects of caffeic acid (and a combination of caffeic acid and metformin) on cervical cancer cell lines. The first article [57] showed that, in the aggressive metastatic human cervical HTB-34 (ATCC-CRL1550) cancer cell line, the exposure to caffeic acid (100 μM) activated AMP-kinase (AMPK), a metabolic sensor with an anti-tumor effect [58,59]. Activated AMPK decreased protein expression of ATP citrate lyase (ACLY), stearoyl-CoA desaturase 1 (SCD1), and fatty acyl-CoA elongase-6 (ELOVL6), enzymes necessary for fatty acid synthesis. The combination of caffeic acid (100 μM) and metformin (10 mM) potentiated these effects [57]. The level of unsaturated fatty acids in HTB-34 cells dropped significantly after exposure. Cancer cells need fatty acids to form new membranes when cells grow. Therefore, fatty acid deprivation in cancer cells inhibits their proliferation. Caffeic acid (100 μM) exposure also decreased the expression of the glucose transporter GLUT1 and increased the activity of mitochondrial pyruvate dehydrogenase, oxidative decarboxylation, and oxidative stress in HTB-34 cells [57]. In the second article [60], caffeic acid (100 μM) decreased the cell viability of metastatic cervical cancer cells (SiHa) but not normal human fibroblasts (FB). Caffeic acid also increased oxidative stress in SiHa cells but not FB cells.

Caffeic acid activates AMPK, which then inhibits acetyl-CoA carboxylase-1 (ACC1) activity and the expression of SREPB1c [57]. Unlike in HTB-34 cells, the exposure increases the protein expression of ATP citrate lyase and fatty acyl-CoA elongase and fails to change the level of lipids in cells [57]. The third article [61] focused on the effect of caffeic acid on epithelial–mesenchyme transition. Losing markers of epithelial cells, e.g., E-cadherin, and gaining mesenchymal phenotype with markers such as vimentin, makes carcinoma cells more aggressive. A typical signaling molecule that promotes epithelial–mesenchyme transition is transforming growth factor beta (TGF-β). Caffeic acid (100 μM) increased the E-cadherin expression and decreased vimentin expression in the human cervical squamous cell line C-4I exposed to TGF-β, and, in this way, it effectively reversed the epithelial–mesenchyme transition. (TGF-β stimulates the epithelial–mesenchyme transition). Caffeic acid also increased mRNA levels of TIMP-1 and TIMP2 (tissue inhibitors of metalloproteinases 1 and 2), and decreased mRNA levels of VEGFA (vascular endothelial growth factor A), metalloproteinases MMP-2, and MMP-9 [57], essential for aggressive tumor growth and metastases [62].

To summarize, in cervical cancer cells, caffeic acid increases the expression of AMPK (Table 1), which then deregulates the expression of enzymes involved in the fatty acid synthesis (Figure 3). Caffeic acid also prevents the epithelial–mesenchyme transition by increasing the expression of E-cadherin and decreasing the expression of vimentin and metalloproteinases (Table 1) (Figure 3).

## 4. Caffeic Acid and Diabetes, Obesity, and Metabolic Syndrome

The Mayo Clinic website describes metabolic syndrome as “a cluster of conditions that occur together, increasing your risk of heart disease, stroke, and type 2 diabetes. These conditions include increased blood pressure, high blood sugar, excess body fat around the waist, and abnormal cholesterol or triglyceride levels”(https://www.mayoclinic.org/diseases-conditions/metabolic-syndrome/symptoms-causes/syc-20351916 accessed on 8 August 2022). Published data show that caffeic acid has a wide range of effects against these conditions.

### 4.1. Diabetes

Castro and coworkers [63] showed that caffeic acid (50 mg/kg) reduced blood glucose levels in streptozocin-induced diabetic mice. They attributed this effect to the ability of caffeic acid to modulate purinergic signaling and, in this way, reduce oxidative stress and act in an anti-inflammatory way. In a similar diabetic model, caffeic acid (35 mg/kg) normalized blood insulin levels and antioxidant parameters: superoxide dismutase (SOD), CAD protein, and glutathione [64]. In alloxan-induced diabetic mice, caffeic acid (50 mg/kg) decreased blood glucose levels, increased hepatic glucokinase (GCK) levels, normalized body weight, and reduced LDL blood levels [65]. Caffeic acid also lowered serum levels of liver enzymes such as alanine transaminase (ALT), aspartate aminotransferase (AST), alkaline phosphatase (ALP), lactate dehydrogenase (LDH), and blood urea, and showed protective and regenerative effects on the kidney and liver. In streptozotocin-induced gestational diabetes in rats, caffeic acid (in a dose-dependent manner) normalized fetus weight, blood lipids, and antioxidant enzymes superoxide dismutase (SOD), glutathione peroxidase (GPX), catalase (CAT), and glutathione negatively altered by diabetes [66]. In insulin-resistant neural cells of high fat diet-induced diabetic rats, caffeic acid increased the expression of the leptin receptor, phospho-JAK2, GLUT3, Akt, and PI3K, and in this way, sensitized cells to insulin signaling [67]. It also increased glucose intake in neural cells. These effects lead authors to suggest that caffeic acid can ameliorate memory function.

In human umbilical vein endothelial cells, HUVECs, caffeic acid (100 μM) inhibited the formation of advanced glycosylation end products, decreased the expression of inflammatory factors interleukin-1β (IL-1β), interleukin-18 (IL-18), and caspase-1, and decreased the production of reactive oxygen species [68]. In the same type of cells, a much lower concentration of caffeic acid (10 nM) improved intracellular redox status and decreased pro-inflammatory NF-κB signaling [69]. In the human stabilized endothelial cell line Ea.hy926, 10 nM caffeic acid showed a similar effect [70]. Additionally, 10 nM caffeic acid decreased apoptosis in Ea.hy926 cells exposed to high glucose. In the context of published data describing the various effects of caffeic acid, the biological activity of caffeic acid at a concentration of 10 nM is remarkable. A higher concentration of caffeic acid (10 μM) also decreased the expression of the receptor for advanced glycation end-products (RAGE) and inflammatory stress marker C-reactive protein (CRP), as well as vascular cell adhesion molecule-1 (VCAM-1), and monocyte chemoattractant protein-1 (MCP-1), in cultured human endothelial cells (HEC) [71].

In mice with chronic stress-induced insulin resistance, caffeic acid (5 and 10 mg/kg)decreased serum levels of glycosylated hemoglobin, tumor necrosis factor-α (TNF-α), and interleukin-1β (IL-1β) [72]. Caffeic acid (various concentrations) also improved oxidative stress in Fe^2+^-induced pancreatic injury: it normalized the level of glutathione, superoxide dismutase (SOD), and catalase (CAT) activity [73].

Approximately 75% of glucose in the blood is cleared by skeletal muscle. To make this possible, glucose transporter GLUT4 must reach the cell membrane. Both insulin signaling and exercise activate the GLUT4 transport while using different signaling pathways. An essential step in the exercise-activated pathway is activating AMPkinase (AMPK). Caffeic acid (100 μM and 1 mM) activated AMPK and its downstream target acetyl-CoA-carboxylase (ACC) in rat skeletal muscle [74]. In this way, caffeic acid helps decrease hyperglycemia if combined with physical exercise.

To summarize, in subjects with diabetes, caffeic acid decreases oxidative stress and inflammation, stimulates insulin sensitivity by inducing PI3K/Akt signaling, prevents damage caused by advanced glycation end-products, and increases the presence of GLUT4 in muscles by activating AMPK.

### 4.2. Obesity

Caffeic acid can also influence fat tissue. Two basic types of adipocytes exist in our bodies: white and brown. The brown adipocytes are more prone to start lipolysis (which leads to losing weight). The reason for this is a higher number of mitochondria in brown adipocytes [75]. Both β3-adrenergic stimulation and cold exposure can activate brown adipocytes and make them start lipolysis to gain energy, while white adipocytes serve more like a passive depot of energy storage. Nevertheless, it is possible to transform white adipocytes into brown ones [75].

Caffeic acid (5 μM, 10 μM, and 50 μM) decreased the expression of key genes of white adipogenic differentiation, including adiponectin, CAAT/enhancer-binding protein alpha (CEBPA), and fatty acid-binding protein 4 (FABP4), and increased the expression of brown adipocyte markers: cell death activator CIDE-A (CIDEA), and uncoupling protein 1 (UCP1) in human Simpson-Golabi-Behmel syndrome (SGB) adipocytes [76]. Caffeic acid also decreased protein expression of PPARγ and lipid accumulation and increased glycerol release [76]. Such results suggest a positive effect of caffeic acid on the “browning” of white adipocytes. Interestingly, a more robust effect was achieved by combining caffeic acid with its derivative chlorogenic acid [76]. In AML12 cells (mouse liver cells), caffeic acid (50 μM) decreased the lipid accumulation and the expression of endoplasmic reticulum stress markers induced by palmitate (250 μM) [77]. It also increased the expression of autophagy markers: microtubule-associated protein 1A/1B light chain 3B (LC3) and autophagy-related 7 (ATG7) [77]. In the differentiated pre-adipocyte cell line 3T3-L1, caffeic acid (31.25 μM and 62.5 μM) significantly reduced lipid content and inhibited intracytoplasmic reactive oxygen species [78]. Caffeic acid (50 μM) also significantly decreased PPARγ protein expression and lipid accumulation in primary-cultured rainbow trout adipocytes [79]. PPARγ represents a major regulator of adipogenesis, especially adipocyte differentiation and lipid accumulation [80]. When the adipocytes were co-exposed to obesitogen rosiglitazone, caffeic acid reversed its effect [79].

To summarize, caffeic acid decreased lipid accumulation and promoted the white-to-brown transition of adipocytes.

### 4.3. Atherosclerosis

One of the major diseases connected with obesity is atherosclerosis. During atherosclerosis development, vascular inflammation plays a significant role. [81].

Caffeic acid (20 μM) showed a significant anti-atherosclerotic effect on human umbilical vein endothelial cells: it decreased interleukin-8 (IL-8) production, toll-like receptor 4 (TLR4) protein expression, and NF-κB signaling induced by the adipokine resistin [82]. In the same type of cells, caffeic acid (25 μM) also inhibited NF-κB-induced expression of adhesion molecules: intracellular adhesion molecule 1 (ICAM-1), vascular adhesion molecule 1 (VCAM-1), and E-selectin [83]. Once expressed on the cell surface, these adhesion molecules are responsible for interactions between blood components and vein endothelial cells [83]. Among others, they facilitate leukocyte adhesion to the endothelium, which represents one of the first steps in atherosclerosis development [84]. In male Wistar rats, caffeic acid (50 mg/kg, p.o.) improved the lipid profile and significantly reduced atherosclerotic lesions [85]. Oxidized LDL represents one of the major risk factors for atherosclerosis, as it causes endothelial dysfunction, an early event in the pathogenesis of cardiovascular diseases [84]. Caffeic acid (100 μM) decreased the activation of endothelial growth factor receptor (EGFR) stimulated by oxidized LDL in ECV-304 endothelial cells and GM-08133A smooth muscle cells [86].

To summarize, caffeic acid decreased pro-inflammatory NF-κB signaling and the expression of adhesive molecules ICAM-1, VCAM-1, and E-selectin in vascular endothelial cells.

## 5. Effects of Caffeic Acid on Brain-Related Diseases

Another pool of published data about caffeic acid describes its effect on brain-related diseases, with most data focusing on counteracting the symptoms of Alzheimer’s disease; a few others describe the effect of caffeic acid on depression or Parkinson’s disease.

### 5.1. Alzheimer’s Disease

The main components of plaques found in the brains of patients with Alzheimer’s disease consist of β-amyloid peptides and tau proteins. The essential step for tau protein aggregation is tau phosphorylation which may also play a role in initiating β-amyloid toxicity. One of the kinases that phosphorylate tau protein is glycogen synthase kinase-3 beta (GSK3β); insulin signaling inhibits the activity of this kinase. Therefore, a hypothesis suggests that GSK3β deregulation in neurons may be a key point in developing Alzheimer’s disease [87].

Feeding hyperinsulinemic rats with caffeic acid (30 mg/kg b.w./day) for 30 weeks significantly improved their memory and learning impairments caused by a high-fat diet [88]. In the brain of hyperinsulinemic rats, caffeic acid normalized superoxide dismutase (SOD) activity and glutathione levels, inhibited glycogen synthase kinase 3β (GSK3β) activity, and decreased the level of β-amyloid and phosphorylated tau protein [88]. Sul and coworkers [89] found similar effects in vitro: the pretreatment with caffeic acid (10 μg/mL) decreased the level of phosphorylated tau protein and GSK3β stimulated by the exposure to 10 μM amyloid-β_25-35_ in rat pheochromocytoma cells PC12. In vitro, caffeic acid (800 μM) prevented the β-amyloid_1-42_ aggregation [90]. It also promoted the disaggregating of mature fibrils in an aqueous solution in the presence of liposomes, which simulated the presence of cell membranes [90]. In the rat model of Alzheimer’s disease established by injection of amyloid-β_1-40_ into the rats, caffeic acid (100 mg/kg for two weeks) significantly improved learning deficits and increased cognitive function (demonstrated by the Morris water maze task). Caffeic acid (100 mg/kg for two weeks) also suppressed oxidative stress, inflammation, NF-κB-p65 protein expression, and caspase-3 activity [91]. In a rat model of Alzheimer’s disease established by intracerebroventricularly administered streptozotocin, caffeic acid (40 mg/kg/day p.o.) showed a similar effect [92]. In an aluminum chloride-induced dementia in rats, caffeic acid (100 mg/kg, p.o.) improved cognitive ability and normalized acetylcholine esterase activity, nitrite and glutathione levels, as well as the protein expression of catalase (CAT) and glutathione-S-transferase (GST) in the brain [93]. In an amyloid-β_25-35_-injected Alzheimer’s disease mouse model, caffeic acid (50 mg/kg/day) improved cognitive functions and inhibited lipid peroxidation and nitric oxide formation in the brain [94]. The majority of people with Alzheimer’s disease suffer from decreased acetylcholine esterase activity and increased butyrylcholine esterase activity [95], and acetylcholinesterase and butyrylesterase inhibitors represent an effective treatment for the disease [96,97]. Caffeic acid (12 μg/mL) inhibited acetylcholinesterase and butyrylcholinesterase activity in the brain of untreated rats in vitro [98]. In acrolein-induced oxidative stress, a situation connected with Alzheimer’s disease [99], caffeic acid (25 μM) protected HT22 mouse hippocampal cells against ROS and glutathione depletion [100]. It also counteracted the disruptive effects of acrolein on p-ERK1/2, p-p38, and p-JNK1 expression [100].

To summarize, in subjects with Alzheimer’s disease, caffeic acid decreases oxidative stress and improves cognitive functions, probably by inhibiting NF-κB and GSK3β signaling and acetylcholinesterase and butyrylcholinesterase activity (Figure 4). Additionally, even though the authors failed to mention it in their papers, we consider the inhibitory effect of caffeic acid on 5-lipoxygenase as another factor in protecting the brain against damage [101,102,103,104].

### 5.2. Depression

In depressed rats, caffeic acid (10 and 30 mg/kg) normalized noradrenalin and tryptophan levels in a dose-dependent manner [105]. Caffeic acid also increased the expression of brain-derived neurotrophic factor (BDNF) in stressed mice; the effect was mediated by 5-lipoxygenase inhibition [106]. BDNF, a neurotrophin that modulates neuroplasticity in the brain, is regularly decreased in depressed patients [107].

### 5.3. Parkinson’s Disease

Protein α-synuclein controls vesicle trafficking in neurons [108]. Its A53T mutated form plays a significant role in developing Parkinson’s disease as its aggregates damage synaptic vesicles, mitochondria, and other cell structures [109]. In A53T α-synuclein transgenic mice, caffeic acid (10 mg/kg) activated the JNK/Bcl-2-mediated autophagy pathway and, in this way, reduced the level of A53T α-synuclein in the substantia nigra of the brain [110].

## 6. Antibacterial and Antiviral Activity of Caffeic Acid

### 6.1. Antibacterial Activity

The antibacterial activity of caffeic acid was tested mostly using *Staphylococcus aureus*, a Gram-positive pathogen able to form biofilms [111]. It is often resistant to antibiotics and disinfectants and, therefore, more difficult to treat [112].

Kwon and coworkers described that caffeic acid (1.0 mg/mL) inhibited the growth of S*taphylococcus aureus* [113]. They hypothesized that caffeic acid inhibited proline dehydrogenase (PRODH), an enzyme necessary for providing energy and managing the redox potential in cells [114]. Caffeic acid (10 mg/mL) also inhibited the secretion of α-hemolysin [115]. *Staphylococcus aureus* secretes α-hemolysin to promote the hemolysis of erythrocytes. α-hemolysin represents one of the major virulence factors of *Staphylococcus aureus* [115]. In the RN-4220 and –1199B resistant strains of *Staphylococcus aureus*, caffeic acid (1024 μg/mL) inhibited the MrsA and NorA efflux pumps responsible for the resistance [112]. Caffeic acid also showed promising inhibitory activity against tetR and tetM efflux pumps in silico*,* which could help fight tet efflux-based tetracycline-resistant bacteria [116]. Caffeic acid (1 mg/mL) inhibited the growth of four clinically significant bacteria: *Escherichia coli*, *Pseudomonas aeruginosa*, *Listeria monocytogenes*, and *Staphylococcus aureus* [117]. Caffeic acid inhibited their replication alone and when combined with Gentamycin, Ciprofloxacin, and Streptomycin [117]. Pinho and coworkers [118] confirmed the effectiveness of caffeic acid (5 mg/mL) against *Staphylococcus aureus, Staphylococcus epidermidis,* and a bit less against *Klebsiella pneumoniae*.

### 6.2. Antiviral Activity

Performing experiments withinfluenza virus A (IFV-A),poliovirus type 1 virus (PV1), and herpes simplex virus 1 (HSV1), Utsunomiya and coworkers [119] showed that caffeic acid (6 mM) inhibited the growth of both DNA and RNA viruses, with RNA viruses being possibly more sensitive. Additionally, the inhibitive effect depended on receiving caffeic acid up to three hours after infection; after that, the effect decreased [119]. Caffeic acid (400 μM) notably inhibited hepatitis C virus (HCV) replication, increased heme oxygenase-1 (HO-1) expression (HO-1 can trigger interferon α antiviral response), and erythroid 2-related factor 2 (Nrf2) expression [120]. In HepG2.2.15 cells, caffeic acid (40 μM) inhibited herpes B virus (HBV) DNA replication; in duck HBV-infected ducklings, caffeic acid (100 mg/kg/day) significantly decreased the level of HBV DNA in serum [121]. In HEp-2 and Vero cells, caffeic acid (8 mM) inhibited the multiplication of HSV1, but only if added early after infection; the addition of caffeic acid six hours after infection showed no effect [122]. Those results suggest that caffeic acid can inhibitHSV-1 multiplication only at the beginning of the process. Langland and coworkers [123] tested the effect of chelates consisting of caffeic acid and metal and non-metal ions against herpes simplex virus 1 (HSV1), herpes simplex virus 2 (HSV2), vaccinia virus (VACV), and a VSV-Ebola pseudo-typed virus. The antiviral activity of caffeic acid increased 100-fold with the addition of Fe^3+^, molybdate and phosphate [123]. Caffeic acid (1mM) also inhibited the growth of severe fever with thrombocytopenia syndrome virus (SFTSV); specifically, it inhibited the binding of the virus to the host cells [124]. In their later work, Ogawa and coworkers showed that the effect against SFTSV depends on the o-dihydroxybenzene backbone of caffeic acid [125].

## 7. Summary

Caffeic acid has shown a wide range of effects beneficial to human health. Its inhibitive effects on cancer cell growth are mediated mainly by inhibiting the PI3K/Akt pathway, MAPK pathway and NF-kB signaling with the consequent inhibition of VEGF. In diabetic rodents, caffeic acid also decreased NF-kB signaling, decreased glucose blood levels, normalized hepatic enzyme levels, improved redox status, and decreased advanced glycation end-products signaling. In adipose tissue, caffeic acid promoted the shift from white adipocytes into brown adipocytes by affecting their differentiation markers. In vein endothelial cells, caffeic acid decreased NF-kB signaling and the expression of adhesive molecules that participates in forming of atherosclerotic plaques. In rodents with Alzheimer’s disease, caffeic acid improved cognitive skills and redox status and decreased the formation of beta-amyloid plaques; the mechanism of these changes correlated with decreased GSK3β levels. In rodents with induced depression, caffeic acid normalized tryptophan and noradrenalin levels; in rodents with Parkinson’s disease, caffeic acid decreased levels of mutated α-synuclein by inducing autophagy. Caffeic also demonstrated antibacterial and antiviral effects: it successfully inhibited the growth of resistant *Staphyloccocus aureus* strains, mostly by inhibiting their efflux pumps. It also inhibited DNA and RNA viruses’ growth as long as it was added at the beginning of the infection.

All these beneficial effects will undoubtedly please coffee lovers. Nevertheless, the question remains whether daily consumption of various beverages suffices to build up caffeic acid blood levels high enough to affect cells.

## Figures and Tables

**Figure 1 ijms-24-00588-f001:**
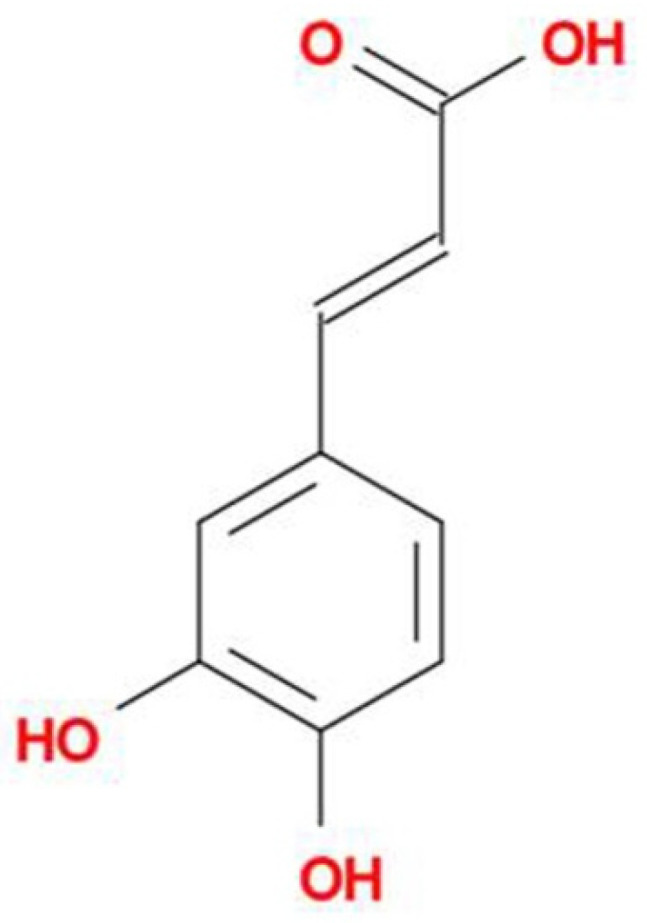
Chemical structure of caffeic acid.

**Figure 2 ijms-24-00588-f002:**
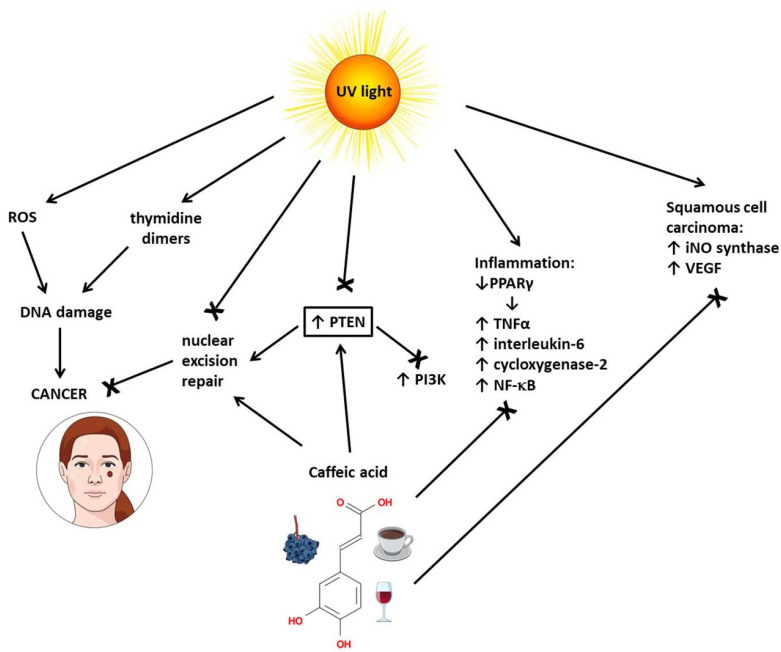
Protective effect of caffeic acid against skin cancer. ROS means reactive oxygen species, PTEN means phosphatase and tensin homolog, PI3K means phosphoinositide 3-kinase, PPARγ means peroxisome proliferator-activated receptor gamma, TNFα means tumor necrosis factor alpha, NF-κB means nuclear factor kappa B, iNO synthase means inducible nitric oxide synthase, and VEGF means vascular endothelial growth factor.

**Figure 3 ijms-24-00588-f003:**
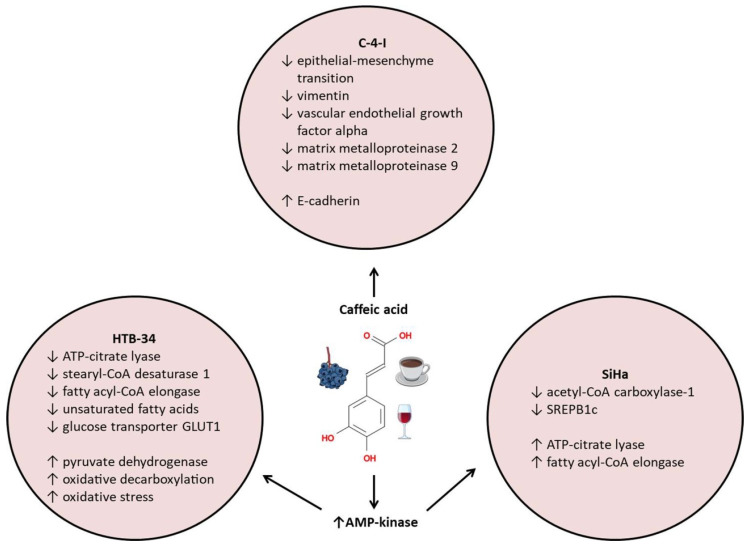
Effects of caffeic acid on metastatic human cervical cancer cells. SREPB1c means sterol regulatory element-binding proteins, and CoA means coenzyme A.

**Figure 4 ijms-24-00588-f004:**
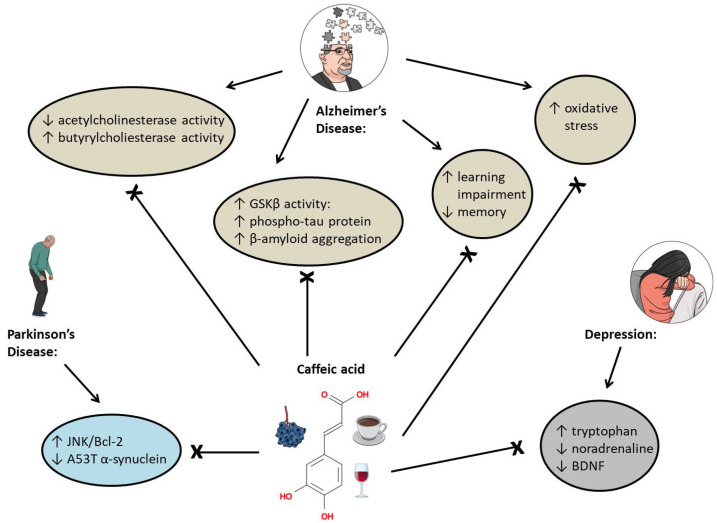
The effects of caffeic acid on brain-related diseases. GSKβ means glycogen synthase kinase 3β, A53T is a type of mutation, and BDNF means brain-derived neurotrophic factor.

**Table 1 ijms-24-00588-t001:** Altered protein and mRNA expression in various types of cancer cells when exposed to caffeic acid.

	Caffeic Acid	
hepatocellular carcinoma		
HepG2, HCC97H	20 μM	↓HIF-1α
HepG2	100 μM	↓NF-κB/IL-6/STAT3
		↓VEGF, MM-9
HepG2, Hep3B, sorafenib-resistant HuH7	20 μM	↓mortalin
breast cancer		
MCF7	50 μM	↓ER, PKB/Akt
		↓IGF-1R
MCF7	171 μg/mL	↑p21 mRNA
		↑MCL1 mRNA
skin cancer		
human dermal fibroblasts and mouse skin	40 μM	↑XPC, XPA, PTEN
		↑TFIIH-p44, ERCC1
squamous cell carcinoma	15 mg/kg	↓iNOS, VEGF
		↑p53
A431, SK-MEL-5, SK-MEL-28	40 μM	↓ERK1/2
HaCaT	100 μM	↓NF-kB/Snail
lung cancer		
H1299 cells and H1299-xenografts	100 μM	↑Bid, Bax
	(with paclitaxel)	↑ cas-3/7, cas-9
		↑p-JNK, p-ERK1/2
A549	100 μM	↑survivin, Bcl-2
LA-795	60 μM	↓p-MEK1/2, p-ERK1/2
		↓cyclin D, vimentin
		↓beta-catenin
		↓TMEM16A
oral cancer		
CAL-27	65 μg/mL	↑p53
		↑PRODH
cervical cancer		
HTB-34 (ATCC-CRL1550)	100 μM	↑AMPK, GLUT1
		↓ACLY, SCD1, ELOVL6
SiHa	100 μM	↑AMPK
		↓ACC1, SREPB1c
		↑ACLY, ELOVL6
C-4I	100 μM	↑E-cadherin
		↓vimentin
		↑TIMP-1 and -2 mRNA

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
