# Peer review of "Caffeic Acid and Diseases—Mechanisms of Action"

_ijms, 2022, doi:10.3390/ijms24010588_

Round 1
Reviewer 1 Report
This is an interesting and nicely written review article that is very timely. I especially like the extensive review of the various target areas and the nice summarization after each section. There are a few sentence/grammar errors as indicated in the following lines:
24, 31,49, 54, 55-56 (what does ortho position really refer to here?), 272-273 and 285-286...why different font size? Line 341: how is 'better' defined? Lines 340-346...appear to have no references here. Lines 445-446...keep figure legend on same page as figure. line 517. In addition, lines 612-625 are references out of alphabetical order.
Reviewer 2 Report
The authors have significantly summarised the mechanisms involved with caffeic acid. The manuscript can be accepted with minor checks of grammar and spelling.
Reviewer 3 Report
1. I would suggest to prepare a table being the summary of caffeic acid role in cancer with a special emphasis on a up/down-regulation of various factors.
2. Since the Author presents mostly in vitro and in vivo results, I was wondering whether there are some publications reporting caffeic acid efficacy in human studies? This would greatly improve the quality of the paper.
3. I would suggest to divide the "brain-related diseases" into neurological and neuropsychiatric ones
4. Please try to characterize well the compound in terms of its structure. This may help to understand the possible mechanism of action.
5. The Author demostrates beneficial effects of the compound. However, there is no information on its possible side effects. Please introduce a subsection demonstrating the above mentioned, if any.
